# Synthesis of Zeolite 4A from Kaolin and Its Adsorption Equilibrium of Carbon Dioxide

**DOI:** 10.3390/ma12091536

**Published:** 2019-05-10

**Authors:** Peng Wang, Qi Sun, Yujiao Zhang, Jun Cao

**Affiliations:** 1College of Materials and Metallurgy, Guizhou University, Guiyang 550025, China; pengwang180907@163.com (P.W.); 13339614162@163.com (J.C.); 2State Key Laboratory of Advanced Processing and Recycling of Nonferrous Metals, Lanzhou University of Technology, Lanzhou 730050, China; zhangyujiao105@163.com

**Keywords:** kaolin, zeolite 4A, carbon dioxide, adsorption equilibrium

## Abstract

Zeolite 4A was successfully prepared by hydrothermal synthesis using low-grade kaolin as a raw material. The properties of the synthesized zeolite 4A were characterized by means of X-ray diffraction (XRD), Fourier transform infrared (FTIR), ^29^Si magic-angle spinning (MAS)-nuclear magnetic resonance (NMR) and ^27^Al MAS-NMR, X-ray fluorescence (XRF), scanning electron microscopy (SEM)-energy-dispersive spectrometry (EDS), transmission electron microscopy (TEM), Brunauer–Emmett–Teller (BET), thermogravimetry (TG)-differential thermal analysis (DTA), and carbon dioxide adsorption analysis. The textural properties of the synthesized zeolite 4A was further studied by BET analysis technique. The thermal stability analysis showed that the heat resistance of the synthesized zeolite 4A is up to 940 °C. In addition, it is found that the Langmuir model has the best agreement with the adsorption equilibrium data for carbon dioxide by synthesized zeolite 4A and commercial zeolite 4A. Meanwhile, the carbon dioxide adsorption analysis confirmed that the maximum equilibrium adsorption amount of carbon dioxide on synthesized zeolite 4A is 59.3820 mL/g, which is higher than the 55.4303 mL/g of the commercial zeolite 4A.

## 1. Introduction

Carbon dioxide is a gas with fierce greenhouse effect. Since the beginning of industrial revolution, more and more carbon dioxide has been released into the atmosphere owing to overexploitation of fossil fuels. Relevant data show that the increasing amount of carbon dioxide in the atmosphere results in global temperature rises by 55% [1,2,3,4]. Carbon dioxide emitted into the atmosphere pollutes the environment and causes an abnormal climate. Meanwhile, carbon dioxide also plays a vital role in the chemical destruction of the ozone layer. Therefore, it seems more urgent to deal with the removal of carbon dioxide. Currently, a variety of methods have been studied to remove carbon dioxide, including chemical adsorption [5], physical adsorption [5], chemical conversion [6], cryogenic separation [7], membrane separation, and so on [8]. Adsorption methods, because of their mild operating conditions, low energy consumption, fast adsorption rate, strong regeneration performance, and stable performance, are already favorable and have been paid more and more attention by researchers at home and abroad. Scholars such as Malek Alaie and Delgado et al. [9,10] showed application of metalorganic frameworks (MOFs), alumina, and zeolite in the removal of carbon dioxide. 

Zeolite with open-framework aluminosilicate structures was [11], and remains, a kind of adsorption material with excellent performance. Compared with other adsorbents, zeolite not only has large specific surface and microporous structure but also has many holes with the same size which can accommodate adsorbed molecules in crystal lattice [6]. In addition, the internal surface of the zeolite is highly polarized and has a strong electrostatic field [6,12]. Meanwhile, carbon dioxide with high polarity quadrupole can react with cations in zeolite [13,14]. Currently, Garshasbi V. et al. [8] reported the carbon dioxide adsorption equilibrium on zeolite 13X prepared from natural clays. Gholipour F. et al. [4] studied the adsorption equilibrium of methane and carbon dioxide using zeolite 13X as the adsorbent material. Therefore, zeolite, a nanoporous inorganic material, has broad application prospects in the field of carbon dioxide removal. However, most of the zeolite used in life is synthesized from chemical raw materials. In this study, zeolite 4A is synthesized by a simple hydrothermal method using low-grade kaolin as a raw material and the adsorption equilibrium of carbon dioxide on zeolite 4A is studied. The synthesis of zeolite 4A by using low-grade kaolin as a raw material reduces the synthesis cost of the zeolite and protects the environment. In addition, the study makes full use of waste resources.

In this study, it is reported that zeolite 4A is synthesized by hydrothermal synthesis using low -grade kaolin as a raw material. The properties of the synthesized zeolite 4A were characterized by X-ray diffraction (XRD), Fourier transform infrared (FTIR), ^29^Si magic-angle spinning (MAS)-nuclear magnetic resonance (NMR) and ^27^Al MAS-NMR, X-ray fluorescence (XRF), scanning electron microscopy (SEM)-energy-dispersive spectrometry (EDS), transmission electron microscopy (TEM), Brunauer–Emmett–Teller (BET), and thermogravimetry (TG)-differential thermal analysis (DTA). Meanwhile, using the static capacity method [12], the adsorption equilibrium isotherm of carbon dioxide on zeolite 4A was studied by adsorption experiment system. In addition, the adsorption equilibrium data of carbon dioxide were used to fit the Langmuir isotherm model and the Freundlich isotherm model, respectively.

## 2. Experimental Section

### 2.1. Materials

The raw kaolin and commercial zeolite 4A used in this study were supplied by Jieao New Building Materials Company (Guiyang, China), located in southwestern part of China. The sodium hydroxide (98%, AR) used in this study was purchased from Shanghai Aladdin Biochemical Technology Co., Ltd. (Shanghai, China) The solution was prepared using deionized water throughout this study. The chemical composition analysis of raw kaolin and metakaolin is shown in Table 1.

### 2.2. Characterization Techniques

The structural feature and crystallinity of the samples were analyzed by German Bruker AXS D8-Focus (Berlin, German). The Fourier transform infrared spectrums of the samples were measured using a PerkinElmer spectrometer (PerkinElmer, Waltham, MA, USA). The chemical composition of the samples were determined on an Axios instrument (PANalytical B.V., Aermoluo, Holland) using wavelength dispersive X-ray fluorescence spectroscopy. Solid state ^27^A1 and ^29^Si MAS-NMR spectra were measured on an Agilent 600M solid nuclear magnetic spectroscopy (Agilent, PaloAlto, CA, USA). A Hitachi SU8010 scanning electron microscope (Hitachi Limited, Tokyo, Japan) was used for the sample microscopic morphology and microarea composition analysis. The TEM images of the samples were measured using a FEI TF20 JOEL 2100F transmission electron microscope (FEI, Hillsboro, AL, USA) with a maximum acceleration voltage of 200 kV. Thermal stability analysis tests were performed using a TG/DTA7300 integrated thermal analyzer (Seiko, Tokyo, Japan). The BET experiment was performed using an ASAP2460 instrument (Micromeritics instrument corp, Georgia, IN, USA). The carbon dioxide adsorption equilibrium of zeolite 4A was measured using a 3H-2000PH1 type carbon dioxide analyzer (Bestech Instrument Technology (Beijing) Co., Ltd., Beijing, China).

### 2.3. Synthesis of Zeolite 4A

First of all, a solution having an alkali concentration of 2.5 mol·L^−1^ was prepared by weighing 10 g of solid sodium hydroxide. Then, 15 g of metakaolin was added to the alkaline solution, and the solution was quickly stirred to make it uniformly mixed. Then, the solution was aged at 60 °C for 4 h and continuously stirred at 300 r·min^−1^. The reaction temperature was raised. The solution was hydrothermally crystallized at 90 °C for 3.5 h and continuously stirred at 500 r·min^−1^. Finally, after the reaction was completed, the solution was filtered, washed, and dried to obtain zeolite 4A. 

### 2.4. Carbon Dioxide Adsorption Equilibrium Experiment

The zeolite was subjected to a degassing treatment under high vacuum at 473 K for 4 h to remove moisture and impurity gases adsorbed on the adsorbent. The system was carefully inspected to ensure that all connections were not leaking and used a vacuum pump to evacuate the system. Then, 2.0 g of zeolite was weighed and loaded onto the adsorption reactor. The pure carbon dioxide gas was introduced into the adsorption device for adsorption measurement. In this process, the upper limit of the test pressure was 45.0 bar and the temperature of the adsorption chamber was 298 K. At this temperature, the adsorption equilibrium is established when the pressure is kept constant, and is used to calculate the adsorption equilibrium amount.

## 3. Results and Discussion

Raw kaolin, a clay composed mainly of kaolinite, contains large amounts of silicon and aluminum sources. However, the silicon and aluminum sources in kaolin are in an inactive state, which makes it difficult to directly react to the synthesis of zeolite 4A. A crucial step is activation treatment of raw kaolin to obtain a more reactive phase, metakaolin [15]. Figure 1 illustrates the thermal analysis curve of raw kaolin. According to the TG curve, the kaolin shows two mass-reduction processes. The first mass-reduction process is attributed to the loss of adsorbed water molecules in the temperature range of 10–400 °C. The second mass-reduction process is in the temperature range of 400–600 °C, which is ascribed to the reorganization of octahedral and tetrahedral layers of the raw kaolin, resulting in the dehydroxylation of the structural OH [15,16,17]. At the same time, an endothermic peak appears in the DTA curve at around 490 °C. The thermal stability analysis of the raw kaolin shows that the transition temperature of kaolin to metakaolin is around 490 °C. The chemical compositions of the raw kaolin, metakaolin, synthesized zeolite 4A, and commercial zeolite 4A could be derived from Table 1. It can be concluded from Table 1 that the major chemical compositions of raw kaolin are SiO_2_, Al_2_O_3_, and a small amount of Fe_2_O_3_. Meanwhile, it is found that the weight percentages of SiO_2_, Al_2_O_3_, and Fe_2_O_3_ in raw kaolin are 55.466, 39.240, and 1.415, respectively. The silicon–aluminum molar ratio in the raw kaolin was increased from 2.4 to 2.5 by calcination treatment at 700 °C for 4 h, and a small amount of Fe_2_O_3_ was removed. Due to the low content of Fe_2_O_3_, it had little effect on the synthetic zeolite and did not require iron removal treatment. The silicon–aluminum molar ratio of 2.5 is in accordance with the requirement of the silicon and aluminum components of zeolite 4A synthesis.

Figure 2A shows the phase analysis of the raw kaolin, metakaolin, synthesized zeolite 4A, and commercial zeolite 4A. It could be observed from Figure 2Aa that the raw kaolin mainly contains kaolinite and a small amount of illite and quartz. However, according to the Joint Committee on Powder Diffraction Standards (JCPDS) [15,18], the feature diffraction peaks of kaolinite occur at 2θ = 12.4° and 24.8°, which is in accordance with the reference value of single-phase kaolinite. Upon calcinations at 700 °C for 4 h, the XRD pattern exhibits the expected significant characteristic, the disappearance of the diffraction peaks of kaolinite, in comparison to the pattern of raw kaolin, accompanied by the appearance of amorphous aluminosilicate, as shown in Figure 2Ab. Meanwhile, characteristic peaks of the impurity illite and quartz are observed in the diffraction peak of metakaolin [15]. Figure 2B shows the FTIR spectrums of raw kaolin, metakaolin, synthesized zeolite 4A, and commercial zeolite 4A. From Figure 2Ba, it could be observed that there are a lot of unambiguous definition FTIR vibration bands in the range of 400–1400 cm^−1^, due to the vibrations of Si-O, Si-O-Al, and Al-OH [19]. The vibration bands at 1031 cm^−1^, 912 cm^−1^, 541 cm^−1^, and 468 cm^−1^ could be assigned to the stretching vibration of Si-O units, the bending vibration of Al-OH units, the bending vibration of the Si-O-Al units, and the bending vibration of Si-O units in the raw kaolin structure, respectively. These vibration bands are removed during the conversion to metakaolin, and new feature vibration bands appear at 1077 cm^−1^, 796 cm^−1^, 694 cm^−1^ and 479 cm^−1^. The loss of the vibration bands at 912 cm^−1^ indicates that the Al-OH units are lost [20]. The reason is that the hydroxyl bond is broken under high temperature calcination. Meanwhile, the feature band of metakaolin is seen at 796 cm^−1^ [21]. In addition, from Figure 2(Ca,b), it could be observed the raw kaolin with hexagonal platy morphology and the appearance of amorphous metakaolin. 

From Figure 2(Ac,d,Bc,d), it could be observed that the XRD pattern and FTIR structure of synthesized zeolite 4A match well with commercial zeolite 4A, which confirms that zeolite 4A is successfully synthesized. For XRD pattern of synthesized zeolite 4A, according to JCPDS [21,22], the feature diffraction peaks of zeolite 4A occur at 2θ = 7.2°, 10.3°, 12.6°, 16.2°, 21.8°, 24.0°, 27.2°, 29.9°, and 34.2°, which is in accordance with the reference value of single-phase zeolite 4A. However, it could be seen from the diffraction pattern that illite and quartz, treated as impurities, still exist. Recent studies have reported the stability of illite and quartz during hydrothermal synthesis [15]. The XRD patterns in Figure 2A correspond to the FTIR spectrums in Figure 2B. For FTIR structure of zeolite 4A, the vibration bands at 1001 cm^−1^ and 471 cm^−1^ could be assigned to the stretching vibration of Si-O or Al-O units and the vibration of Si-O-Al units in the zeolite 4A structure, respectively. The vibration bands appeared at 668 cm^−1^ and 556 cm^−1^ could be assigned to the vibration modes of the zeolite 4A framework [22]. The SEM morphology of synthesized zeolite 4A was studied as shown in Figure 2(Cc,d). It could be clearly seen that the zeolite 4A have a typical cubic morphology with homogeneous size distribution. SEM morphology of commercial zeolite 4A is presented in Figure 3. The morphology of synthesized zeolite 4A matches well with the commercial zeolite 4A in terms of size and structure. The solid state ^27^A1 and ^29^Si MAS-NMR spectra of synthesized zeolite 4A are shown in Figure 4. It could be seen from Figure 4a that only one resonance at around 57.810 ppm was exhibited, which implies that Al species dissolved from metakaolin entered the 4-coordinate Si-O network to form the framework Si(OAl)3(OSi) in the alkaline solution [23,24,25]. In addition, the full width at half maximum (FWHM) of resonance around 57.810 ppm was narrow, which illustrates that the crystallinity of the zeolite 4A is favorable [23]. The typical ^29^Si MAS-NMR resonance for zeolite 4A, around −92.017 ppm, was quite sharp, indicating that zeolite 4A has favorable lattice order as shown in Figure 4b [24]. In addition, EDS spectrums of raw kaolin, metakaolin, synthesized zeolite 4A, and commercial zeolite 4A is presented in Figure 5. For the raw kaolin and metakaolin, it could be clearly seen the peaks constituting the elements O, Al, and Si in the spectrum. For zeolite 4A, a new peak associated with the element Na appears in the spectrum. Combined with Table 1, it can be revealed that positively charged sodium ions are present in the synthesized zeolite 4A as equilibrium cations. 

Following that, TEM technique was applied to further study the formation of synthesized zeolite 4A as shown in Figure 6. It could be seen from Figure 6a,b that the zeolite 4A with good crystallinity grows solidly. The lattice fringes, as shown in Figure 6c,d, indicate pore size of 0.4–0.5 nm. The electron diffraction pattern as shown in Figure 6e indicates the formation of body-centered cubic crystals. As could be seen from Figure 6f, elemental analysis spectrum shows that the main components of the zeolite 4A are Na, Al, Si, and O [26]. The percentages of Na, Al, Si, and O are 5.2, 28.41, 29.55, and 36.93%, respectively. In addition, selected-area electron diffraction patterns obtained from the edges of the zeolite 4A are presented in Figure 7. It could be seen from Figure 7b that there are the spotty diffraction rings, which indicates the polycrystallinity of the zeolite 4A structure [11]. From Figure 8, it could be confirmed again that the elements Na, Si, Al, and O are present in the zeolite 4A structure. In addition, these mappings clearly show that the zeolite 4A contains a large amount of silica and alumina [26]. From Figure 8b, it could be seen that the Na, Si, Al, and O elements are uniformly distributed in zeolite 4A.

The nitrogen adsorption-desorption isotherm of the synthesized zeolite 4A and pore size distribution of the synthesized zeolite 4A are presented in Figure 9. It could be seen from Figure 9a that there is a hysteresis loop, indicating the existence of mesoporosity [19]. The pore size distribution of the zeolite 4A obtained by the Barrett–Joyner–Halenda equation (BJH) is shown in Figure 9b. From Figure 9b, it could be concluded that the BJH adsorption average pore diameter and BJH desorption average pore diameter are 10.5063 nm and 9.0646 nm, respectively. The micropore volume, BET surface area, and micropore surface area of the synthesized zeolite 4A and commercial zeolite 4A are listed in Table 2. From Table 2, the micropore volume, BET surface area and micropore surface area of synthesized zeolite are significantly higher than commercial zeolite 4A.

Thermal stability of synthesized zeolite 4A during heat treatment was studied as shown in Figure 10. From the TG curve, the two weight reductions of zeolite 4A could be observed. The first weight reduction of zeolite 4A occurred in the temperature range of 20−180 °C. At the same time, it could be seen from the DTA curve that the strongest endothermic peak was near 140 °C. The main reason was that the water molecules adsorbed on the zeolite 4A were completely removed. The second weight reduction occurred in the temperature range between 180 and 500 °C. The main reason is that extraframework water molecules were removed [27]. However, a weak endothermic peak at about 930 °C could be seen from the DTA curve. It is known from the TG curve that it has nothing to do with weight reduction. The main reason is the collapse of the zeolite 4A structure [19,25]. Most importantly, we could conclude that the heat resistance of zeolite 4A is as high as 940 °C.

A comparative study of carbon dioxide adsorption on synthesized zeolite 4A and commercial zeolite 4A at 298 K is shown in Figure 11. It could be seen from Figure 11a that in the low pressure range, the carbon dioxide adsorption amount increases rapidly with the increase of the equilibrium pressure, and the carbon dioxide adsorption amount tends to be stable when the pressure reaches 45 bar [28]. Meanwhile, it could be concluded that the synthesized zeolite 4A adsorbs more carbon dioxide than the commercial zeolite 4A in the pressure range of 0–45 bar. Combined with Table 2, the reason is that synthesized zeolite 4A has a larger micropore surface area and micropore volume compared to commercial zeolite 4A [2]. The maximum adsorption amount of carbon dioxide in the adsorption curve indicates that the zeolite 4A reaches the saturation adsorption limit at 298 K. Meanwhile, it is shown that all active sites on the zeolite 4A were sufficiently bound to the carbon dioxide gas. From Figure 11b, it could be clearly seen that the maximum adsorption amounts of synthesized zeolite 4A and commercial zeolite 4A for carbon dioxide are 59.382 mL/g and 55.4303 mL/g, respectively. In this study, a nonlinear regression analysis was performed to determine the adsorption isotherm model of the zeolite, as shown in Figure 11c,d. The typical type I behavior was shown according to the IUPAC adsorption isotherm classification criteria [2], which demonstrates the monolayer adsorption mechanism of microporous adsorbents [28,29]. Combined with Table 3, the Langmuir isotherm model fully complies with the isotherm curve for the adsorption of carbon dioxide by synthesized zeolite 4A and commercial zeolite 4A, which indicates the monolayer adsorption on the surface of the uniform adsorbent, and the interaction between the adsorbed particles are negligible [30]. Meanwhile, it can be seen from Figure 11c,d that a large number of adsorbed active sites are activated within a suitable equilibrium pressure range. In addition, the study has a guiding significance for the determination of adsorption pressure and regeneration pressure as the adsorption pressure changes during the adsorption process.

## 4. Conclusions

Zeolite 4A was successfully synthesized from low-grade kaolin by using a hydrothermal method. The BET analysis indicated that the micropore volume, BET surface area, and micro pore surface area of the zeolite 4A were 0.001141 cm^3^·g^−1^, 13.3723 m^2^·g^−1^, and 2.4624 m^2^·g^−1^, respectively. The thermal stability analysis test showed that the heat resistance of the synthesized zeolite 4A is up to 940 °C. In addition, it was found that the Langmuir model has the best agreement with the adsorption equilibrium data for carbon dioxide by synthesized zeolite 4A and commercial zeolite 4A. Meanwhile, it was also found from the carbon dioxide adsorption analysis that the maximum equilibrium adsorption amount of carbon dioxide on synthesized zeolite 4A is 59.3820 mL/g, which is higher than the 55.4303 mL/g of the commercial zeolite 4A. Further studies on the mechanism of how the zeolite 4A adsorbs carbon dioxide are under investigation in our lab at present.

## Figures and Tables

**Figure 1 materials-12-01536-f001:**
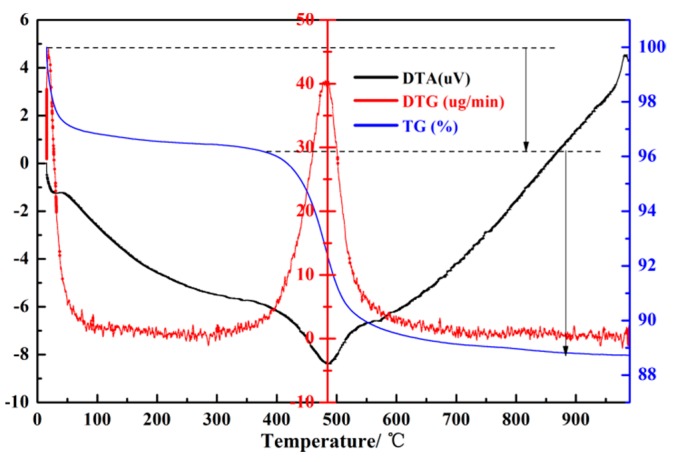
Thermal analysis of raw kaolin.

**Figure 2 materials-12-01536-f002:**
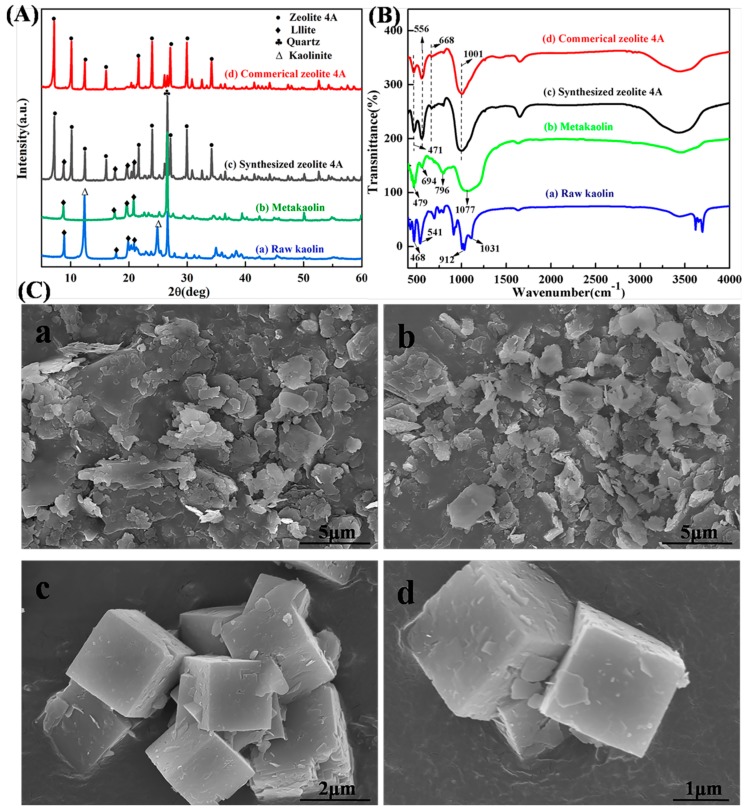
Comparison of the raw kaolin, metakaolin, synthesized zeolite 4A, and commercial zeolite 4A, in (**A**) X-ray diffraction (XRD) patterns, (**B**) Fourier transform infrared (FTIR) spectra, and (**C**) scanning electron microscopy (SEM) morphology: (**a**) raw kaolin, (**b**) metakaolin, and (**c**,**d**) synthesized zeolite 4A.

**Figure 3 materials-12-01536-f003:**
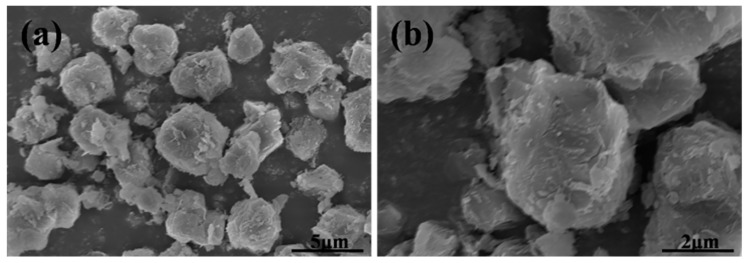
(**a**,**b**) SEM morphology of commercial zeolite 4A.

**Figure 4 materials-12-01536-f004:**
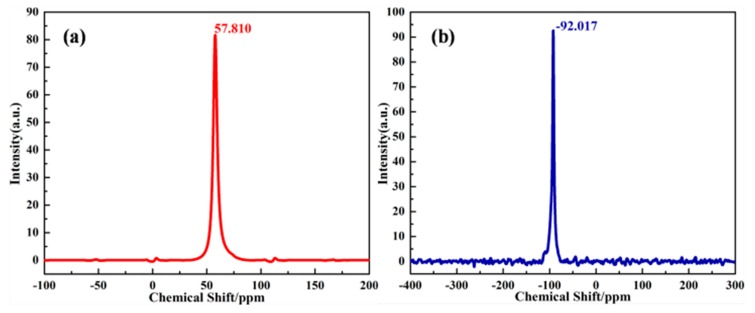
(**a**) ^27^Al magic-angle spinning (MAS)-nuclear magnetic resonance (NMR) and (**b**) ^29^Si MAS-NMR spectra of synthesized zeolite 4A.

**Figure 5 materials-12-01536-f005:**
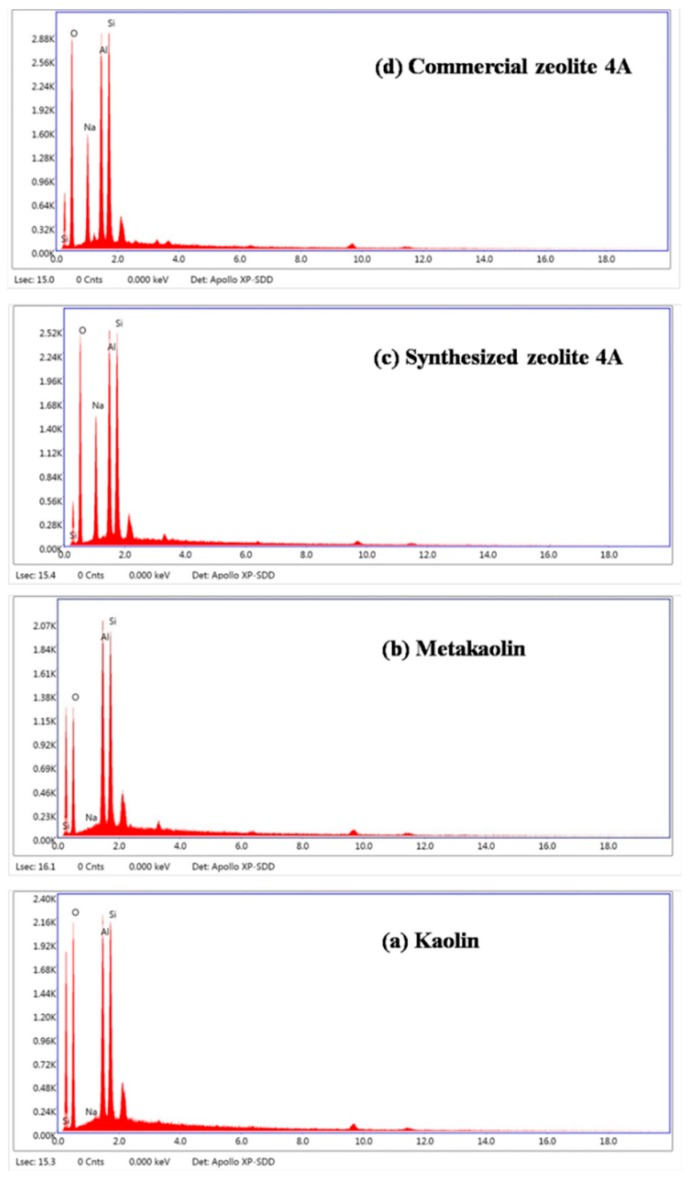
Energy-dispersive spectrometry (EDS) spectrums of (**a**) raw kaolin, (**b**) metakaolin, (**c**) synthesized zeolite 4A, and (**d**) commercial zeolite 4A.

**Figure 6 materials-12-01536-f006:**
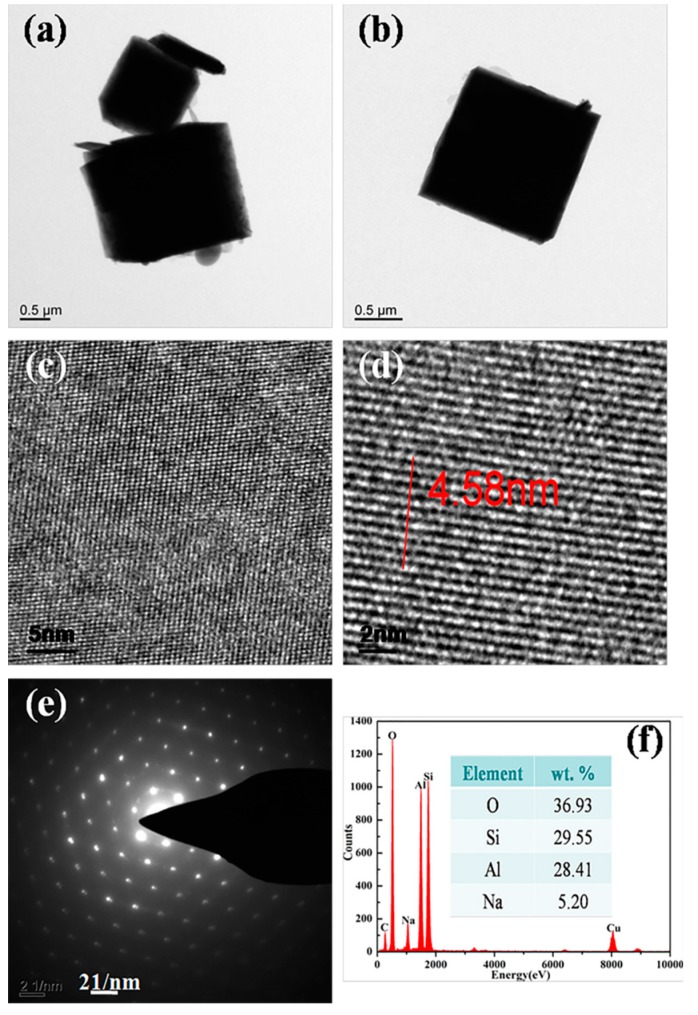
Transmission electron microscopy (TEM) results of synthesized zeolite 4A: (**a**,**b**) TEM micrograph, (**c**,**d**) lattice fringes, (**e**) selected-area electron diffraction (SAED), and (**f**) energy-dispersive X-ray (EDX) pattern.

**Figure 7 materials-12-01536-f007:**
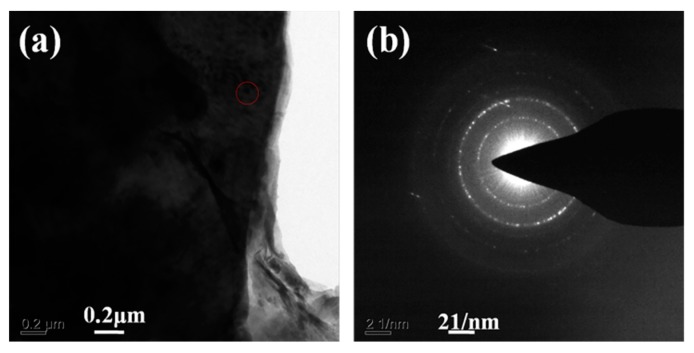
(**a**) TEM image of the synthesized zeolite 4A and (**b**) the spotty diffraction rings of the synthesized zeolite 4A.

**Figure 8 materials-12-01536-f008:**
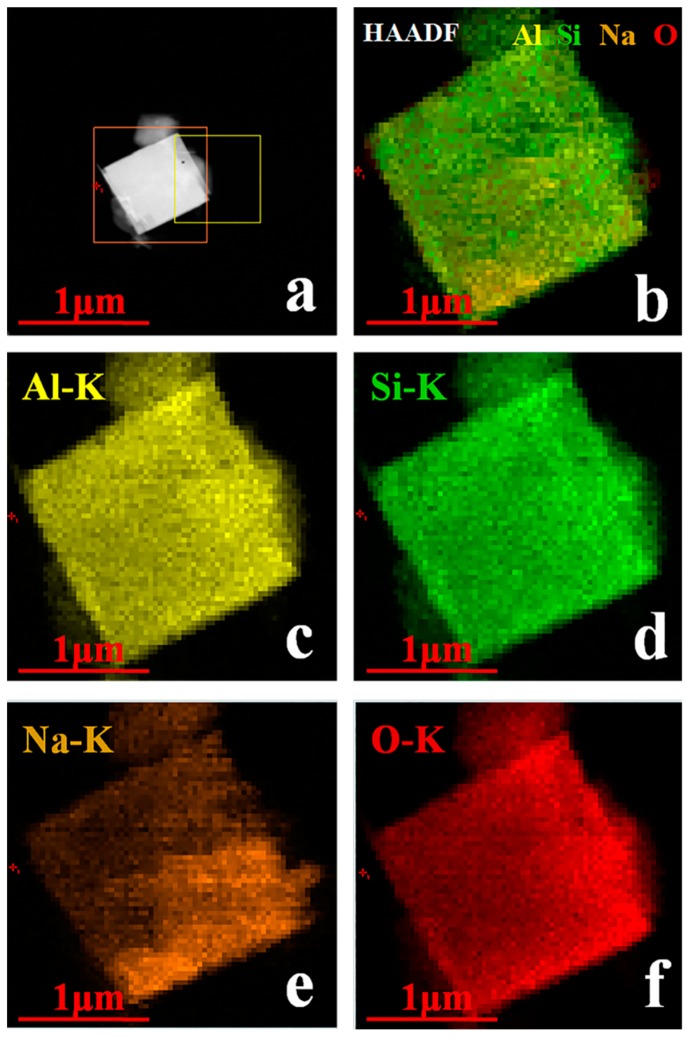
TEM, high-angle annular dark-field (HAADF)-scanning transmission electron microscopy (STEM) elemental mapping of synthesized zeolite 4A. (**a**,**b**) The region where examined mapping, (**c**) Al element distribution, (**d**) Si element distribution, (**e**) Na element distribution, (**f**) O element distribution.

**Figure 9 materials-12-01536-f009:**
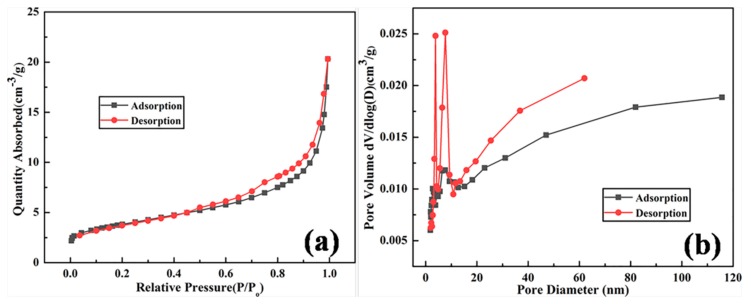
(**a**) The N_2_ adsorption-desorption isotherm of synthesized zeolite 4A and (**b**) the pore size distribution of synthesized zeolite 4A.

**Figure 10 materials-12-01536-f010:**
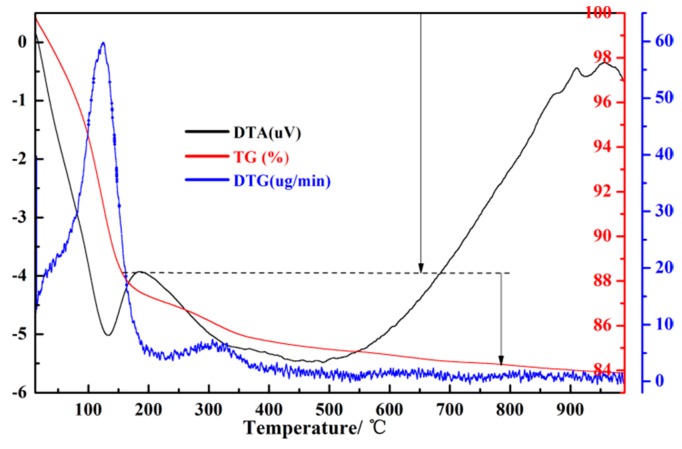
Thermal analysis of synthesized zeolite 4A.

**Figure 11 materials-12-01536-f011:**
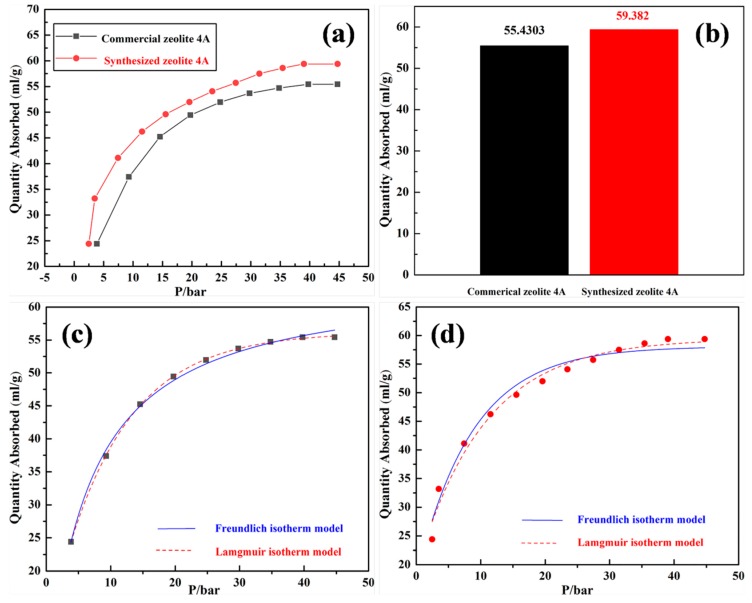
(**a**) A comparative study of carbon dioxide adsorption on synthesized zeolite 4A and commercial zeolite 4A at 298 K. (**b**) The histogram of maximum adsorption amount for carbon dioxide: synthesized zeolite 4A (red histogram); commercial zeolite 4A (black histogram). Adsorption isotherm of carbon dioxide on (**c**) commercial zeolite 4A and (**d**) synthesized zeolite 4A at 298 K: Freundlich equation (solid line); Langmuir equation (dotted line).

**Table 1 materials-12-01536-t001:** The chemical composition analysis of raw kaolin, metakaolin, synthesized zeolite 4A, and commercial zeolite 4A.

Formula (wt %)	Raw Kaolin	Metakaolin	Synthesized Zeolite 4A	Commercial Zeolite 4A
SiO_2_	55.466	56.268	46.730	46.220
Al_2_O_3_	39.240	38.735	33.135	29.720
Fe_2_O_3_	1.415	1.339	1.0849	2.090
Na_2_O	0.054	0.069	16.350	15.740
SiO_2_/Al_2_O_3_	2.4	2.5	2.4	2.6

**Table 2 materials-12-01536-t002:** The textural properties of synthesized zeolite 4A and commercial zeolite 4A.

Samples	BET Surface Area (m^2^/g)	Micropore Surface Area (m^2^/g)	Micropore Volume (cm^3^/g)
Synthesized zeolite 4A	13.3723	2.4624	0.001141
Commercial zeolite 4A	7.3738	0.6201	0.000227

**Table 3 materials-12-01536-t003:** The correlation coefficient values for the Langmuir model and Freundlich model at 298 K.

Adsorbents	Correlation Coefficient (R^2^)	
	Langmuir Isotherm Model	Freundlich Isotherm Model
Synthesized zeolite 4A	0.99921	0.97560
Commercial zeolite 4A	0.99958	0.97034

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
