# Peer review of "Synthesis of Zeolite 4A from Kaolin and Its Adsorption Equilibrium of Carbon Dioxide"

_materials, 2019, doi:10.3390/ma12091536_

Reviewer 1 Report

This is a sound paper describing the Synthesis of zeolite 4A from kaolin and its adsorption  equilibrium of carbon dioxide by a combined XRD, 11 FTIR, 29Si MAS-NMR and 27Al MAS-NMR, XRF, SEM-EDS, TEM, BET, TG-DTA approach. The manuscript is well written, only minor revision are suggested (see attached manuscript).

Author Response

Question 1: the phase

Response 1: Thanks for the reviewer’s suggestion and the authors believe that the opinion of reviewer is very important to our work. Therefore, the authors have revised it in paragraph 5 of the manuscript and highlighted in red.

The structural feature and crystallinity of the samples were analyzed by German Bruker AXS D8-Focus.

Question 2: German Broker AXS D8-Focus

Response 2: Thanks for the reviewer’s suggestion and the authors are very sorry for the letter mistake. Therefore, the authors have revised it in paragraph 5 of the manuscript and highlighted in red.

The structural feature and crystallinity of the samples were analyzed by German Bruker AXS D8-Focus.

Question 3: please to specify

Response 3: Thanks for the reviewer’s suggestion, and the authors think they are very useful for our work. Therefore, the authors have described in detail in paragraph 6 of the manuscript and highlighted in red.

First of all, a solution having an alkali concentration of 2.5 mol L-1 was prepared by weighing 10 g of solid sodium hydroxide. 15 g of metakaolin was added to the alkaline solution, and the solution was quickly stirred to make it uniformly mixed. Then, the solution was aged at 60 ℃ for 4 h and continuously stirred at 300 r min-1. The reaction temperature was raised. The solution was hydrothermally crystallized at 90 for 3.5 h and continuously stirred at 500 r min-1. Finally, after the reaction was completed, the solution was filtered, washed and dried to obtain zeolite 4A. 

Question 4: Other phases are present. This is not pure zeolite 4A.

Response 4: Thanks for the reviewer’s suggestion and the authors believe that the opinion of reviewer is very important to our work. Therefore, the authors have revised it in paragraph 10 of the manuscript and highlighted in red.

From Figure 2Ac,d and Figure 2Bc,d, it could be observed that XRD pattern and FTIR structure of synthesized zeolite 4A match well with commercial zeolite 4A, which confirms that zeolite 4A is successfully synthesized. 

Question 5: In the temperature range 180-500°C the weight loss can be ascribed to extraframework water molecules

Response 5: Thanks for the reviewer’s suggestion and the authors believe that the opinion of reviewer is very important to our work. Therefore, the authors have revised it in paragraph 13 of the manuscript and highlighted in red.

The second weight reduction occurs in the temperature range between 180 and 500. The main reason is that extraframework water molecules are removed [28].

Reviewer 2 Report

The authors synthesized Zeolite 4A from low-grade Kaolin, and it is tested for CO2 capture. The textural properties of synthesized Zeolite 4A were analyzed by XRD, FTIR, 29Si MAS-NMR and 27Al MAS-NMR, XRF, SEM-EDS, TEM, BET, TG-DTA.

1.     This paper needs extensive language and format editing.

2.     Lines 17-19; 247-248: The CO2 capture capacity of the synthesized zeolite is only 7% higher than the commercial zeolite. I think the usage of ‘much higher’ is improper here. It is misleading.

3.     Line 32: The reference citation has to be sequential.

4.     The introduction is not well written. The literature review of CO2 adsorption on zeolites is missing in the introduction section. There should be some comparison between this work and the research reported in the literature.

5.     Lines 63-75: All the characterization techniques used in the work should be described in detail. For example: The experimental conditions used in measuring the CO2 adsorption equilibrium were missing.

6.     Figure 2 and Figure 3: The FTIR and XRD features are not clearly shown in Figure 2. Separate Figure 2 C (a-d) and combine with Figure 3.  

7.     Lines 126-129: The IR features are not explained in detail here. Which IR features represent Si-O Vibrations? It would be easy to understand if the IR bands are tabulated.

8.     The description/discussion of the figures should be sequential. For example: line 150: Figure 4 was described ahead of Figure 2C and Figure 3

9.     Line 193 Table 3: The textural properties of the synthesized zeolite and commercial zeolite should be compared.

10.  The results and discussion section is poorly organized and is very difficult to follow.

11.  Line 216: How is the adsorption capacity calculated? It should be discussed in experimental section

12.  Line 219: It should be 'synthesized zeolite' not 'synthetic zeolite'.

13.  The morphology of the synthesized zeolite is same as the commercial zeolite and the CO2 adsorption capacity is slightly higher than that of commercial zeolite. What is the novelty of this study? Is the synthesized zeolite thermally more stable than the commercial zeolite or is it cheaper than the commercial zeolite? The authors need to clarify on the objective of this study in the introduction section.

Author Response

Question 1: This paper needs extensive language and format editing.

Response 1: Thanks for the reviewer’s suggestion and the authors are very sorry for these grammar mistakes and format editing errors. According to your request, the English throughout the manuscript has been checked many times.

Question 2: Lines 17-19; 247-248: The CO2 capture capacity of the synthesized zeolite is only 7% higher than the commercial zeolite. I think the usage of ‘much higher’ is improper here. It is misleading.

Response 2: Thanks for the reviewer’s suggestion and the authors believe that the opinion of reviewer is very important to our work. Therefore, the authors have revised it in manuscript and highlighted in red.

Line 17-19: Meanwhile, the carbon dioxide adsorption analysis confirmed that the maximum equilibrium adsorption amount of carbon dioxide on synthesized zeolite 4A is 59.3820 ml/g, which is higher than the 55.4303 ml/g of the commercial zeolite 4A.

Line 271-273: Meanwhile, it is also found from the carbon dioxide adsorption analysis that the maximum equilibrium adsorption amount of carbon dioxide on synthesized zeolite 4A is 59.3820 ml/g, which is higher than the 55.4303 ml/g of the commercial zeolite 4A.

Question 3: Line 32: The reference citation has to be sequential.

Response 3: Thanks for the reviewer’s suggestion and the authors believe that the opinion of reviewer is very important to our work. Therefore, the authors have revised it in manuscript and highlighted in red.

Line 29-31: Currently, a variety of methods have been studied to remove carbon dioxide, including chemical adsorption [5], physical adsorption [5], chemical conversion [6], cryogenic separation [11], membrane separation and so on [12].

Question 4: The introduction is not well written. The literature review of CO2 adsorption on zeolites is missing in the introduction section. There should be some comparison between this work and the research reported in the literature.

Response 4: Thanks for the reviewer’s suggestion and the authors believe that the opinion of reviewer is very important to our work. Therefore, the comparison between this work and the research reported in the literature has been discussed in manuscript and highlighted in red.

Line 42-51: Currently, Garshasbi V. et al. [12] reported the carbon dioxide adsorption equilibrium on zeolite 13X prepared from natural clays. Gholipour F. et al. [22] studied the adsorption equilibrium of methane and carbon dioxide using zeolite 13X as the adsorbent material. Therefore, zeolite, a nanoporous inorganic material, has broad application prospects in the field of carbon dioxide removal. However, most of the zeolite used in life is synthesized from chemical raw materials. In this study, zeolite 4A is synthesized by a simple hydrothermal method using low-grade kaolin as a raw material and the adsorption equilibrium of carbon dioxide on zeolite 4A is studied. The synthesis of zeolite 4A by using low-grade kaolin as a raw material reduces the synthesis cost of the zeolite and protects the environment. In addition, the study makes full use of waste resources.

Question 5: Lines 63-75: All the characterization techniques used in the work should be described in detail. For example: The experimental conditions used in measuring the CO2 adsorption equilibrium were missing.

Response 5: Thanks for the reviewer’s suggestion and the authors believe that the opinion of reviewer is very important to our work. Therefore, the authors have revised it in manuscript and highlighted in red.

Line 67-79: The structural feature and crystallinity of the samples were analyzed by German Bruker AXS D8-Focus. The Fourier transform infrared spectrums of the samples were measured using a Perkin-Elmer spectrometer. The chemical compositions of the samples were determined on an Axios instrument using wavelength dispersive X-ray fluorescence spectroscopy. Solid state 27A1 and 29Si MAS NMR spectra were measured on an Agilent 600M solid nuclear magnetic spectroscopy. A Hitachi SU8010 scanning electron microscope was used for the sample microscopic morphology and micro-area composition analysis. The TEM images of the samples were measured using a FEI TF20 JOEL 2100F transmission electron microscope with a maximum acceleration voltage of 200 kV. Thermal stability analysis tests were performed using a TG/DTA7300 integrated thermal analyzer (Seiko, Japan). The BET experiment was performed using an ASAP2460 instrument. The carbon dioxide adsorption equilibrium of zeolite 4A was measured using a 3H-2000PH1 type carbon dioxide analyzer (Bestech Instrument Technology (Beijing) Co., Ltd., China).

Line 88-95: The zeolite was subjected to a degassing treatment under high vacuum at 473 K for 4 h to remove moisture and impurity gases adsorbed on the adsorbent. Carefully inspect the system to ensure that all connections are not leaking and use a vacuum pump to evacuate the system. 2.0 g of zeolite was weighed and loaded onto the adsorption reactor. The pure carbon dioxide gas is introduced into the adsorption device for adsorption measurement. In this process, the upper limit of the test pressure is 45.0 bar and the temperature of the adsorption chamber is 298 K. At this temperature, the adsorption equilibrium is established when the pressure is kept constant, and is used to calculate the adsorption equilibrium amount.

Question 6: Figure 2 and Figure 3: The FTIR and XRD features are not clearly shown in Figure 2. Separate Figure 2 C (a-d) and combine with Figure 3.

Response 6: Thanks for the reviewer’s suggestion and the authors are very sorry about the problems in the paper. Meanwhile, the authors believe that the opinion of reviewer is very important to our work. However, according to your request, the authors believe that the entire structure of the paper must undergo major changes. From the perspective of the revision time of the paper, the author thinks that the work is a bit difficult. Therefore, the authors are very sorry.

Question 7: Lines 126-129: The IR features are not explained in detail here. Which IR features represent Si-O Vibrations? It would be easy to understand if the IR bands are tabulated.

Response 7: Thanks for the reviewer’s suggestion and the authors believe that the opinion of reviewer is very important to our work. Therefore, the IR features have been described in detail in manuscript and highlighted in red.

Line 88-95: The vibration bands at at 1031 cm-1, 912 cm-1, 541 cm-1, and 468 cm-1 could be assigned to the stretching vibration of Si-O units, the bending vibration of Al-OH units, the bending vibration of the Si-O-Al units, and the bending vibration of Si-O units in the raw kaolin structure, respectively.

Question 8: The description/discussion of the figures should be sequential. For example: line 150: Figure 4 was described ahead of Figure 2C and Figure 3.

Response 8: Thanks for the reviewer’s suggestion and the authors believe that the opinion of reviewer is very important to our work. Therefore, the consistency of the order/discussion of the figures in manuscript has been slightly adjusted.

Question 9: Line 193 Table 3: The textural properties of the synthesized zeolite and commercial zeolite should be compared.

Response 9: Thanks for the reviewer’s suggestion, and the authors believe that the opinion of reviewer is very important to our work. Therefore, the textural properties of commercial zeolite have been added in Table 2.

Question 10: The results and discussion section is poorly organized and is very difficult to follow.

Response 10: Thanks for the reviewer’s suggestion and the authors believe that the opinion of reviewer is very important to our work. Therefore, the authors have carefully revised the results and discussion section of the manuscript.

Question 11: Line 216: How is the adsorption capacity calculated? It should be discussed in experimental section

Response 11: Thanks for the reviewer’s suggestion and the authors believe that the opinion of reviewer is very important to our work. Therefore, the carbon dioxide adsorption equilibrium experiment has been added in manuscript and highlighted in red.

Line 88-95: The zeolite was subjected to a degassing treatment under high vacuum at 473 K for 4 h to remove moisture and impurity gases adsorbed on the adsorbent. Carefully inspect the system to ensure that all connections are not leaking and use a vacuum pump to evacuate the system. 2.0 g of zeolite was weighed and loaded onto the adsorption reactor. The pure carbon dioxide gas is introduced into the adsorption device for adsorption measurement. In this process, the upper limit of the test pressure is 45.0 bar and the temperature of the adsorption chamber is 298 K. At this temperature, the adsorption equilibrium is established when the pressure is kept constant, and is used to calculate the adsorption equilibrium amount.

Question 12: Line 219: It should be 'synthesized zeolite' not 'synthetic zeolite'.

Response 12: Thanks for the reviewer’s suggestion and the authors believe that the opinion of reviewer is very important to our work. Therefore, the authors have revised it in manuscript and highlighted in red.

Line 242-244: From Figure 11b, it could be clearly seen that the maximum adsorption amounts of synthesized zeolite 4A and commercial zeolite 4A for carbon dioxide are 59.382 ml/g and 55.4303 ml/g, respectively. 

Question 13: The morphology of the synthesized zeolite is same as the commercial zeolite and the CO2 adsorption capacity is slightly higher than that of commercial zeolite. What is the novelty of this study? Is the synthesized zeolite thermally more stable than the commercial zeolite or is it cheaper than the commercial zeolite? The authors need to clarify on the objective of this study in the introduction section.

Response 13: Thanks for the reviewer’s suggestion. The novelty of the manuscript are as follows:
Currently, commercial zeolite 4A is synthesized from chemical raw materials. However, in this study, zeolite 4A is synthesized by a simple hydrothermal method using low-grade kaolin as a raw material. The synthesis of zeolite 4A by using low-grade kaolin as a raw material reduces the synthesis cost of the zeolite and protects the environment. In addition, the study makes full use of waste resources.

Line 46-51: However, most of the zeolite used in life is synthesized from chemical raw materials. In this study, zeolite 4A is synthesized by a simple hydrothermal method using low-grade kaolin as a raw material and the adsorption equilibrium of carbon dioxide on zeolite 4A is studied. The synthesis of zeolite 4A by using low-grade kaolin as a raw material reduces the synthesis cost of the zeolite and protects the environment. In addition, the study makes full use of waste resources.

Reviewer 3 Report

The manuscript entitled  “Synthesis of zeolite 4A from kaolin and its adsorption equilibrium of carbon
dioxide” presents the synthesis of zeolite 4A from kaolin via hydrothermal method. The properties of
the synthesized zeolite 4A were characterized by means of many methods and this material was used as carbon dioxide adsorbent.  The article is interesting, the characteristics of synthesized zeolite 4A are
well described. The part of manuscript concerning  procedure of synthesis of zeolite 4A and adsorption of carbon dioxide is too short and requires more details. I  recommend this manuscript to publication with major revision. My comments are following: 1. Chapter 2.3. Synthesis of zeolite 4A – this procedure should be described more detailed –
concentration of alkaline solution, time of mixing and aging of mixture containing metakaolin
and NaOH. 2. The authors should add a chapter describing adsorption experiments - conditions for conducting experiments, apparatus, etc. in Part 2. Experimental 3. Table 2 collect only the elemental composition of studied materials and does not give any other
physico-chemical properties of these materials. This table gives practically the same information
as Table 1, only in a different way. 4. Table 4 contains only the values of correlation coefficients for Langmuir and Freundlich models.
I did not found the isotherm parameters, like Freundlich and Langmuir constants, 1/n  and q (maximum adsorption capacity) parameters.  The description of this table does not correspond to its c
ontents. 5. There are a few letter mistakes in the text – line 163 – “ For zeoilite…”, Fig. 11c – “Lamgmuir
isotherm model”

Author Response

Question 1: Chapter 2.3. Synthesis of zeolite 4A-this procedure should be described more detailed-concentration of alkaline solution, time of mixing and aging of mixture containing metakaolin and NaOH.

Response 1: Thanks for the reviewer’s suggestion, and the authors think they are very useful for our work. Therefore, the synthesis process of zeolite 4A has been described in detail in paragraph 6 of the manuscript and highlighted in red.

First of all, a solution having an alkali concentration of 2.5 mol L-1 was prepared by weighing 10 g of solid sodium hydroxide. 15 g of metakaolin was added to the alkaline solution, and the solution was quickly stirred to make it uniformly mixed. Then, the solution was aged at 60 ℃ for 4 h and continuously stirred at 300 r min-1. The reaction temperature was raised. The solution was hydrothermally crystallized at 90 for 3.5 h and continuously stirred at 500 r min-1. Finally, after the reaction was completed, the solution was filtered, washed and dried to obtain zeolite 4A. 

Question 2: The authors should add a chapter describing adsorption experiments-conditions for conducting experiments, apparatus, etc. in Part 2. Experimental

Response 2: Thanks for the reviewer’s suggestion, and the authors think they are very useful for our work. Therefore, the carbon dioxide adsorption equilibrium experiment has been added in paragraph 7 of the manuscript and highlighted in red.

2.4. Carbon dioxide adsorption equilibrium experiment

The zeolite was subjected to a degassing treatment under high vacuum at 473 K for 4 h to remove moisture and impurity gases adsorbed on the adsorbent. Carefully inspect the system to ensure that all connections are not leaking and use a vacuum pump to evacuate the system. 2.0 g of zeolite was weighed and loaded onto the adsorption reactor. The pure carbon dioxide gas is introduced into the adsorption device for adsorption measurement. In this process, the upper limit of the test pressure is 45.0 bar and the temperature of the adsorption chamber is 298 K. At this temperature, the adsorption equilibrium is established when the pressure is kept constant, and is used to calculate the adsorption equilibrium amount.

Question 3: Table 2 collect only the elemental composition of studied materials and does not give any other physico-chemical properties of these materials. This table gives practically the same information as Table 1, only in a different way.

Response 3: Thanks for the reviewer’s suggestion and the authors believe that the opinion of reviewer is very important to our work. Therefore, the authors have deleted Table 2 in manuscript.

Question 4: Table 4 contains only the values of correlation coefficients for Langmuir and Freundlich models. I did not find the isotherm parameters, like Freundlich and Langmuir constants, 1/n and q (maximum adsorption capacity) parameters. The description of this table does not correspond to its contents.

Response 4: Thanks for the reviewer’s suggestion and the authors are very sorry that the description of the table in the manuscript does not correspond to its contents. Therefore, the authors have re-described the table in the manuscript.

Question 5: There are a few letter mistakes in the text -line 163- “ For zeoilite…”, Fig. 11c- “Lamgmuir isotherm model”

Response 5: Thanks for the reviewer’s suggestion and the authors are very sorry for these letter mistakes. The authors have corrected these letter mistakes and the English throughout the manuscript has been checked many times.

 Round  2

Reviewer 2 Report

The authors showed some improvement. 

Reviewer 3 Report

Manuscript  can be published in the present form.